# Genomic Factors and Therapeutic Approaches in HIV-Associated Neurocognitive Disorders: A Comprehensive Review

**DOI:** 10.3390/ijms241814364

**Published:** 2023-09-21

**Authors:** Ana Borrajo, Daniel Pérez-Rodríguez, Carlos Fernández-Pereira, José María Prieto-González, Roberto Carlos Agís-Balboa

**Affiliations:** 1Department of Microbiology and Parasitology, Faculty of Pharmacy, Complutense University of Madrid, 28040 Madrid, Spain; 2Department of Experimental Medicine and Surgery, University of Rome Tor Vergata, 00133 Roma, Italy; 3NeuroEpigenetics Lab, Health Research Institute of Santiago de Compostela (IDIS), Santiago University Hospital Complex, 15706 Santiago de Compostela, Spain; daniel.prz.rodriguez@gmail.com (D.P.-R.); carlosfernandezpereira@gmail.com (C.F.-P.); josemaoscar.prieto@usc.es (J.M.P.-G.); 4Facultade de Bioloxía, Universidade de Vigo (UVigo), Campus Universitario Lagoas-Marcosende, s/n, 36310 Vigo, Spain; 5Translational Neuroscience Group, Galicia Sur Health Research Institute (IIS Galicia Sur), Area Sanitaria de Vigo-Hospital Álvaro Cunqueiro, SERGAS-UVIGO, CIBERSAM-ISCIII, 36213 Vigo, Spain; 6Rare Disease and Pediatric Medicine Group, Galicia Sur Health Research Institute (IIS Galicia Sur), SERGAS-UVIGO, 36312 Vigo, Spain; 7Translational Research in Neurological Diseases (ITEN), Health Research Institute of Santiago de Compostela (IDIS), Santiago University Hospital Complex, 15706 Santiago de Compostela, Spain; 8Servicio de Neurología, Hospital Clínico Universitario de Santiago, 15706 Santiago de Compostela, Spain

**Keywords:** HIV-1, microglia, HIV-1 Associated Neurocognitive Disorders, cART

## Abstract

HIV-associated neurocognitive disorders (HANDs) still persist despite improved life expectancy, reduced viral loads, and decreased infection severity. The number of patients affected by HANDs ranges from (30 to 50) % of HIV-infected individuals. The pathological mechanisms contributing to HANDs and the most serious manifestation of the disease, HIV-associated dementia (HAD), are not yet well understood. Evidence suggests that these mechanisms are likely multifactorial, producing neurocognitive complications involving disorders such as neurogenesis, autophagy, neuroinflammation, and mitochondrial dysfunction. Over the years, multiple pharmacological approaches with specific mechanisms of action acting upon distinct targets have been approved. Although these therapies are effective in reducing viral loading to undetectable levels, they also present some disadvantages such as common side effects, the need for administration with a very high frequency, and the possibility of drug resistance. Genetic studies on HANDs provide insights into the biological pathways and mechanisms that contribute to cognitive impairment in people living with HIV-1. Furthermore, they also help identify genetic variants that increase susceptibility to HANDs and can be used to tailor treatment approaches for HIV-1 patients. Identification of the genetic markers associated with disease progression can help clinicians predict which individuals require more aggressive management and by understanding the genetic basis of the disorder, it will be possible to develop targeted therapies to mitigate cognitive impairment. The main goal of this review is to provide details on the epidemiological data currently available and to summarise the genetic (specifically, the genetic makeup of the immune system), transcriptomic, and epigenetic studies available on HANDs to date. In addition, we address the potential pharmacological therapeutic strategies currently being investigated. This will provide valuable information that can guide clinical care, drug development, and our overall understanding of these diseases.

## 1. Introduction

### 1.1. Background and the Prevalence of HIV-Associated Neurocognitive Disorders

HIV-1 (human immunodeficiency virus-1) infection is a complex and progressive viral illness that affects the immune system. The common modes of transmission of HIV-1 infection include unprotected sexual intercourse, sharing needles, receiving contaminated blood products, and from mother to child during childbirth or breastfeeding [1]. Once the virus enters the body, HIV-1 attaches to the surface of CD4 cells using its glycoprotein spikes, specifically targeting the chemokine receptors on the cell surface. After attachment, HIV-1 fuses with the CD4 cell membrane and releases its genetic material into the cell. The machinery of the host cell then begins to produce new HIV-1 particles that can now infect other CD4 cells or immune cells, perpetuating the cycle of infection [1]. Importantly, HIV-1 can evade the immune system response (production of antibodies against HIV-1 and activation of immune cells to target infected cells) and establish a persistent infection.

After the initial acute infection, many patients enter a stage known as clinical latency or chronic infection. During this stage, the virus continues to replicate at low levels but individuals may not experience noticeable symptoms. This stage can last for years, but over time, the virus gradually damages the immune system. If left untreated, HIV-1 infection can lead to AIDS (acquired immunodeficiency syndrome), which is characterised by severe immune deficiency. At this stage, the immune system is severely compromised and individuals become susceptible to opportunistic infections and serious diseases [1].

In the general population, mental disorders have a greater impact than other maternal, neonatal, nutritional, and non-communicable diseases, including HIV-1, the development of HIV-associated neurocognitive disorders (HANDs), and injuries [1]. Indeed, the lifespans of patients with mental and neurological disorders can be shortened by approximately 15 to 20 years. The incidence of mental health disorders peaks in young adulthood and increases in late adolescence. Of note, mental health disorders significantly contribute to the increased risk of HIV-1 acquisition among various populations. In comparison to the healthy population, individuals with serious mental illness tend to exhibit lower levels of sexual activity. In fact, people with serious mental illnesses usually display irregular sexual behaviours such as erratic condom use, multiple sexual partners, and alcohol and illegal drug consumption before sex [2]. The risk of HIV-1 infection may also increase with the severity of the psychiatric disease.

HANDs involve different types of cognitive impairments, including impaired attention span, memory [3], processing speed, and executive function [2,3]. These difficulties, along with feelings of apathy and depression, often also affect workplace functioning and medication adherence and may cause motor deficits in some cases [2,3]. Research has revealed that patients with HIV-1 suffer mental health disorders at a much higher rate than the general population. Indeed, the results of a multi-site study of more than 2800 HIV-1 participants conducted in the United States of America (USA) found that 36% had major depression and 15.8% had generalised anxiety disorder [3], compared with only 6.7% and 2.1% of the general population, respectively [4]. 

A study from 2018 revealed that 32% and 21% of people with HIV-1 from the southeast of the USA had severe mood and anxiety disorders, respectively [3]. In Canada, the records indicate that 41% of patients living with HIV-1 have a mental health condition compared to 22% of non-HIV-infected adults [5]. Furthermore, a study of patients with HIV-1 in India indicated that 59% had signs of major depression [6]. Similarly, in China, recent data demonstrated a prevalence of the symptoms of depression in 61% of people with HIV-1 [7]. In 2019, a study in Spain revealed that 63% of patients with HIV-1 exhibited symptoms of depression, anxiety, and severe mood or mental disorders [3]. 

Mental health problems can also obstruct the prevention of HIV infection, including regular HIV testing and adherence to pre-expose prophylaxis (PrEP), which is highly effective at protecting individuals from acquiring HIV when taken consistently [3]. The success of combination antiretroviral therapy (cART) improves the quality of life for HIV-infected patients. However, this improvement is being undermined by the growing risk of HANDs, which impacts the overall health of individuals living with this disease.

The current understanding is that HIV enters the central nervous system (CNS) during infection and remains there, replicating in immune cells in the brain. This ongoing replication, combined with the immune response and inflammation triggered by the host, is believed to contribute to the development of HANDs. Another factor that increases susceptibility to HANDs is divergence in host genetics, which can partially explain the varying effectiveness of the human immune response. Extensive research conducted in recent years has emphasised the role of this response, along with inflammation, and various nervous system cells in HANDs. However, further investigation will still be necessary to understand the exact molecular mechanisms responsible for its pathogenesis. Thus, the intricate nature of HAND phenotypes stems from a range of factors, including the continued discovery of new therapeutic targets, research complexity, considerable variability of the virus and changes in the timing of its action, diagnostic uncertainties, and growing intersection between chronic immune activation and comorbidities [8,9,10].

### 1.2. Diagnosis and Comorbidities

There are many approaches to the diagnosis of a HAND, although the Frascati criteria are the most universally employed to identify patterns of cognitive dysfunctions and possible deterioration. To diagnose HANDs according to the Frascati criteria, the impairment must affect a minimum of two cognitive domains. These cognitive areas are detected by administering a neuropsychological battery that includes at least five cognitive domains (executive functions, fine motor or sensory perception skills, learning, attention, and working memory). Thus, for a diagnosis of asymptomatic neurocognitive impairment (ANI), which is characterised by cognitive damage, at least two cognitive domains that do not interfere with daily activities must be identified by the Frascati criteria. Mild neurocognitive disorders (MNDs) represent cognitive damage of at least two cognitive domains to a degree that produces at least mild interference in everyday function [11,12]. Finally, a diagnosis of HIV-associated dementia (HAD) requires the identification of cognitive impairment involving substantial interference and marked incompatibility with daily functioning in at least two cognitive areas, together with severe functional damage [12] (Figure 1).

HIV therapy failure is a common problem in developing countries that increases the risk of the progression of neurocognitive disorders [3]. In this sense, nearly 58% of the burden of dementia occurs in low- and middle-income countries [9]. Many of the affected patients do not receive adequate pharmacological therapy, leading to significant cognitive and psychiatric issues that greatly diminish the quality of life for individuals with HAD [8,12]. In addition, even if these patients received proper therapy plans, these treatments are often ineffective in certain populations. Thus, further genetic studies would still be necessary to try to offer personalised treatments for HIV-infected individuals at risk. Furthermore, we would have to carry out (1) longitudinal studies to understand the principal phenotypes of different HANDs in HIV-infected subpopulations with a very high risk of being adversely affected; (2) studies with biological samples and tissues to correlate neuropathological changes in different brain regions in order to search for genes involved in affected individuals (changes at the RNA, epigenomic, and/or protein levels); and (3) genome-wide association studies (GWAS) utilising more sensitive HAND sub-phenotypes.

The National Institutes of Health (NIH) Brain Initiative and Human Connectome Project, along with studies on biological systems, transcriptomic and epigenetic data analysis, and machine learning methods are all promising approaches to identifying possible biological processes and mechanisms that can be intervened or potential gene networks underlying HANDs, especially HAD [12]. Moreover, a significant association between HAD and mental illnesses such as depression, schizophrenia, and bipolar disorder has been found. These comorbidities are often observed in people living with chronic diseases, including HIV+ patients, and are strongly associated with neuronal deficits and medical non-adherence, impaired functional status, neurocognitive damage, and mortality [6,7]. HANDs strongly affect the functioning of individuals. For example, HAD patients suffer from lower mood and energy levels and decreased activity during depressive events. They may also experience a lack of concentration with increased fatigue and decreased interest in all activities, disturbed sleep patterns, and a diminished appetite. In addition to all the aforementioned indicators, in the mild form there is also usually a clear absence of self-confidence and a tendency towards low self-esteem.

As a result, patients with a HAND, predominantly those with HAD, may be at high risk of improper use of their HIV treatments because they tend to be inconsistent with their medications or may overuse them [7]. Well-established screening tools for depression that can be repeatedly used for assessment are available to clinicians [8]. Moreover, the neurocognitive assessment for HANDs generally includes an evaluation for depression because cognitive impairment is common in these patients and can sometimes be improved with therapeutic interventions including treatment with antiretroviral therapy [8].

Psychiatric disorders such as schizophrenia or bipolar disorder are frequently found in individuals living with HIV who abuse substances or come from disadvantaged social and economic backgrounds. These factors significantly contribute to their heightened susceptibility to neurocognitive diseases [6,8]. Additionally, it is worth noting that a majority of these patients have advanced HIV disease because of the delayed initiation of treatment. Early identification of depression and HANDs is crucial given that they are linked to higher rates of non-adherence to medication as well as other negative outcomes, including increased risky behaviour.

Even mild depression can have deleterious consequences on the lives of HIV-1-infected individuals. It can lead to inconsistent treatment adherence, poor engagement with caregivers, and ultimately, to more serious outcomes [13]. Therefore, the early detection of depression among patients with HIV-1 should help prevent HANDs, especially the most serious forms of the disease. Hence, the timely diagnosis of HANDs should be an integral part of the primary care necessary for good management of the disease. The presence of untreated mental disorders in patients living with HIV-1 strongly impacts their health outcomes, thereby impeding successful treatment and affecting their quality of life. It is also important to study depression and neurocognitive disorders more rigorously by comparing HIV+ and HIV− cohorts of patients with mental health issues. However, these screening tools require extensive staff training and time to for their administration which could limit their use because these kinds of instruments are normally used as part of routine HIV care plans.

The average worldwide prevalence of HANDs in 2022 was 50.41% (95% confidence interval [CI]: 45.56, 55.26), which was similar in Europe at 50.015% in the same year. In turn, in Africa, Asia, and the USA, it was 49.57%, 52.03%, and 50.41%, respectively in 2022. Interestingly, the prevalence of HANDs in studies that used the International HIV Dementia Scale (IHDS) was 36.88% and 59.97% at the IHDS cutoff points <9.5 and <10, respectively. In contrast, according to the global dementia scale (GDS), the estimated global average of HANDs was 40.766%. In comparison, the prevalence of HANDs in cross-sectional, cohort, and case–control studies was 49.52%, 54.08%, and 44.45%, respectively. Sociodemographic variables such as low education levels, older age, clinical and HIV-related variables, advanced disease stage, CD4 counts of 500 cells/dL or less, and psychological variables including comorbid depression all increase the risk of HANDs [13].

The importance of phenotype in HANDs lies in the role that variations in genetic factors play in determining the observable characteristics and clinical outcomes of individuals with this condition (as discussed in more detail in the Section 2). In this sense, genetic variations can contribute to differences in how individuals respond to HIV-1 infection and how it impacts the brain. Thus, by identifying specific genetic factors associated with certain phenotypic traits in HANDs, clinicians can develop more personalised and effective treatment strategies. In addition, some genetic variations are associated with a higher risk of developing a HAND. Therefore, by studying phenotypes and their correlation with specific genetic markers, researchers can identify individuals who are more likely to experience cognitive impairment, thereby allowing for early intervention and targeted monitoring. 

Phenotypes are also useful because they are a valuable source of biomarkers for HANDs that correlate with specific clinical manifestations that can be utilised as indicators for disease progression, treatment response, and prognosis. Moreover, genetic variations can provide insights into the underlying biological mechanisms contributing to HANDs. Thus, by studying how specific genetic factors influence phenotype, researchers can uncover pathways and processes involved in the development of cognitive impairment.

This review describes findings from studies investigating the role of human host genetic factors in the pathogenesis of HAD (the most severe form of HAND). These consist of diverse molecular mechanisms as potential risk factors for developing the infection. We also provide a comprehensive overview of the key pathological processes associated with HANDs, especially emphasising knowledge related to HAD, and explore potential therapeutic approaches targeting relevant signalling pathways that could help address this disease.

## 2. Important Genes Involved in the Development of HIV-Associated Neurocognitive Disorders

There is a close, bidirectional relationship between cognitive impairment and the risk factors associated with HANDs that should be considered when trying to reduce the risk of disease onset. These risk factors are both environmental and genomic and include drug and tobacco consumption, sedentary lifestyles, and unhealthy diets, all of which can strongly impact the appearance of HANDs and should be further studied (Figure 2). Moreover, specific genes are also known to be important in the development of HANDs. In the following discussion, we summarise the most relevant genetic factors involved in this disease (as summarised in Table 1 at the end of this section).

### 2.1. Genes Involved in Inflammation

Over the years, lack of virological control, which is common in patients with HIV because of its late presentation and lack of patient retention in care units, are two factors that contribute to neuronal injury, neuroinflammation, and neurocognitive degeneration [14], with HANDs (especially HAD) being a complication of these. Despite the current availability of improved antiretroviral therapies, HAD is an important concern among HIV-infected individuals. The pervasiveness of these types of inflammation has been attributed to multiple factors, including elevated plasma levels of soluble CD14 (sCD14), soluble CD163 (sCD163), macrophage chemo-attractant protein-1 (MCP-1), and neopterin, among others [13,15]. In addition, some genes have been linked, albeit inconsistently, to HAD, including tumour necrosis factor-α (*TNF-α*), *MCP-1*/*CCL2*, macrophage inflammatory protein 1-α (*MIP-1α*/*CCL3*), stromal cell-derived factor-1 (*SDF1*), HLA alleles, and mannose-binding lectin 2 (*MBL2*) [6]. Additionally, damaged M1 pro-inflammatory microglial activation is considered one of the most unique features of HAD neuropathogenesis and it has been suggested that circulating HIV-1 transactivator protein (Tat) can play a critical role in this process [15].

In in vitro experiments on BV-2 microglial cells, Silveira et al. found that an HIV-1 Tat-induced M1 inflammatory state could be limited by the pharmacological inhibition of the unfolded protein response (UPR)-related protein kinase R-like endoplasmic reticulum kinase. This inhibition reduced overexpression pro-inflammatory mediators such as inducible nitric oxide synthase (iNOS), surface CD16/32, secreted tumour necrosis factor-α (TNF-α), IL-6, monocyte chemoattractant protein (MCP)-1, and nitric oxide (NO), suggesting that pharmacological blockade may partly prevent neuroinflammation and neuronal loss in HAD [16]. 

Other recent work has shown that monocyte-associated cerebrospinal fluid (CSF) biomarkers, such as TNF-α, MCP-1, neopterin, soluble CD14, and neurofilament light chain protein (NFL), are strongly associated with neuronal damage in people both with and without HIV-1 [17,18]. Furthermore, other authors [18,19] have discussed the use of brain tissue, CSF, and blood biopsies to assess glial cell activation or elevated cytokines, as well as the use of tissue samples to indirectly measure neuroinflammation. Indeed, some pro-inflammatory markers present in CSF have been used as indicators of neuroinflammation, given that CSF is directly connected both to the brain [19], inflammatory cytokines, and interleukin-15 (IL-15) [19]. However, there are some concerns about both the variability in reported CSF biomarker concentrations, meaning that the samples may not accurately reflect the state of cells present in the brain, as well as the invasiveness of the procedure.

Blood biopsies are a less invasive methodology, but these assays represent a less direct measure of inflammatory processes in the brain because the exchange of molecules across the blood–brain barrier (BBB) is limited. However, some studies employing blood samples used the presence of inflammatory markers such as C-reactive protein (CRP, a widely-studied cytokine which represents acute inflammation) and cytoskeletal NFL (released into CSF and blood as the result of neuroaxonal impairment) in the diagnosis of neurological diseases [20]. These studies suggested that NFL levels in both CSF and plasma correlate with levels of well-known markers of neuroinflammation such as C-X-C motif chemokine ligand 10 (CXCL10), neopterin [21], the kynurenine-to-tryptophan ratio, and kynurenine metabolites, which are important mediators in neuroinflammation [22]. Thus, these researchers concluded that blood serum NFL may be used as an indirect biomarker of neuroinflammation.

Despite the advantages of directly measuring gene and protein biomarkers, epigenetic signatures represent a more stable and longitudinal measurement of the chronic inflammatory state of these markers. Work has already been carried out to analyse these biomarkers using data from epigenome-wide association studies as a proxy of CRP [23]. However, DNA methylation signatures for NFL remain an unexplored aspect of the study of neuroinflammation in humans. In addition, members of the HLA complex are part of the system involved in HIV disease progression and the development of HAD. This is because HIV-infected cells present viral antigens through HLA I to cytotoxic T lymphocytes (CTLs), also known as CD8 T cells, allowing them to expand, recognise, and attack infected cells [24]. However, helper T cells (CD4 T cells), which are activated by HLA II, are required for an effective CD8 T cell response [25].

HLA class I (HLA-A, HLA-B, and HLA-C) code for cell surface molecules that present antigens formed by the host cell itself, including viral proteins if the cell is infected. In turn, HLA class II genes (HLA-DR, HLA-DP, and HLA-DQ) code for proteins that present viral antigens engulfed by immune cells [24]. HLA I is important in the immunity process of viral cells, while iHLA alleles (particularly HLA class I alleles) affect HIV disease progression. Of note, a previous study of a cohort in the USA showed that the HLA-DR*04 allele was associated with an increased risk of HAD [24]. Moreover, the HLA-A*24 genotype was associated with increased progression of CNS disease. Both these studies concurred with previous work associating HLA alleles with the progression of HIV infection [25].

The expression of genes encoding HLA class I is significantly increased in response to interferon type I and II in HIV encephalitis (HIVE) [26]. HLA-E was also implicated in patients with HAD and HIVE [27], as was HLA-B [28]. Subsequent work postulated that the HLA-DR genotype did not significantly correlate with neurocognitive functions (assessed as a global clinical score, the GCR) or by using histopathological markers in adults with HIV-1 [27]. It would be interesting to study the impact of the HLA gene variants in HANDs at every level of severity in greater depth, because different results have indicated that HLA alleles play a clear role in the progression of HIV infection and may be associated with cognitive impairment [25]. However, it is worth noting that research on the role of HLA genes is limited by the large sample size required for HLA genotyping.

Variations in the MBL2 allele have been previously related to HAD [26], probably because it plays a role in macrophage-related innate and adaptive immunity. In this line, Singh et al. found that children who were homozygous for the B, C, and D alleles (rs1800450, rs1800451, and rs5030737, respectively) suffered greater cognitive impairment [29]. These results were supported by another study in a cohort of Chinese adults which concluded that the same genotypes were associated with mild neurocognitive impairment and HAD [30]. In contrast, subsequent work was unable to replicate these data in patients with HIV-1 with neurocognitive impairment (excluding HAD), with an association between the ‘O’ MBL2 genotype and neurocognitive damage being shown only the HIV-1 control group [31], indicating that there was no relationship with HIV comorbidity [32]. MBL2 levels are increased in the neurons of patients with HIVE [31], but this relationship was not associated with neurocognitive functioning. Although associations between MBL2 single nucleotide polymorphisms (SNPs) were analysed in each GWAS, no significant results were found [33]. Indeed, no specific role in neurocognitive impairment in the context of HIV-1 infection has been shown for MBL2, although further research is still required in this regard.

### 2.2. Genes Related to Serotonin Neurotransmission

Serotonin signalling plays a crucial role in neurobiology because this neurotransmitter regulates various functions in the CNS including mood, sleep–wake cycles, appetite, and pain perception, among others [34]. Given that HANDs are a spectrum of neurological impairments that can occur in individuals living with HIV-1, serotonin signalling has been implicated in the potential pathogenesis of these neurological deficits. Of note, HIV-1 can directly infect the CNS and cause inflammation, leading to neuronal damage. Additionally, the virus can affect serotonin receptors and transporters, disrupting normal serotonin signalling. Changes in serotonin levels and signalling in the brain may contribute to cognitive dysfunction, mood disorders, and other neurological complications associated with HIV-1 infection [34]. Thus, understanding these mechanisms is crucial for developing targeted therapies to mitigate the impact of HIV-1 on the CNS.

In this context, SNPs of tryptophan hydroxylase 2 (*TPH2*) and galactose mutarotase (*GALM*), two serotonin-related genes, have been associated with neurocognitive impairments in HIV-1 patients, especially in the African male population [35]. First, the homozygous TT genotype at SNP rs4570625 was associated with slower executive activity in HIV+ people. According to the Frascati criteria, most of these patients had ANI and some had MND, but none had HAD [12] (see Figure 1). This SNP is located in the transcriptional region of *TPH2*, the rate-limiting enzyme for serotonin biosynthesis, and affects its transcription rate [35]. Second, this study also found a relationship between the rs6741892 polymorphism of *GALM*, which indirectly increases the local release of serotonin and produces memory problems in affected individuals [35]. However, the SLC6A4 5-HTTLPR-region polymorphism did not appear to significantly affect serotonin transport. These data agree with previous studies showing associations between the same SNPs and more limited activity and memory performance [35]. Nonetheless, there were some shortcomings to this work because of its cross-sectional design and lack of a healthy control group.

### 2.3. Involvement of Genes Related to Neurocognitive Disorders in Immune Regulation

To date, studies on the most promising genes thought to be involved in neurocognitive disorders have investigated single nucleotide polymorphisms or other variations in genes involved in (1) the immune system such as C-C chemokine receptor type 5 (*CCR5*), C-C chemokine receptor type 2 (*CCR2*), C-C chemokine ligands type 2, 3, and 5 (*CCL2*, *CCL3*, and *CCL5*, respectively), and members of the HLA complex; (2) lipid metabolism, such as Apolipoprotein Eε4 (*ApoEε4*), as an important risk factor for Alzheimer disease (AD) [35,36]; or (3) the dopaminergic system, such as dopamine active transporter (*DAT*), dopamine receptor genes D2 and D3 (*DRD2* and *DRD3*, respectively), catechol-O-methyltransferase (*COMT*), and brain-derived neurotrophic factor (*BDNF*) [37].

The CCR5Δ32 deletion or a SNP in the HIV co-receptor can produce protection against neurocognitive damage in HAD or HAND [38]. Furthermore, the genotypes of *CCR2* and that of one of the natural ligands of CCR5, *CCL3*, are significantly associated with CSF neuroinflammatory markers and worse cognitive functioning in HIV-associated neurocognitive disease [38]. It is important to clarify that DAT is a protein responsible for the reuptake of dopamine from the synaptic cleft back into the presynaptic neuron after signal transmission. Thus, DAT plays a critical role in terminating dopamine signalling and maintaining proper neurotransmission. Therefore, dysregulation of DAT function can lead to altered dopamine levels in the brain, which may contribute to HANDs [37].

The functions of *DRD2* and *DRD3* are also very important in the context of HANDs. These dopamine signalling pathway receptors and are involved in regulating various functions including motor control, reward, mood, and cognition [37]. Dysfunctions in these receptors have been implicated in several neuropsychiatric disorders and alterations in their expression or signalling could be relevant in the context of HAND. *COMT* is another enzyme that plays a role in the breakdown and inactivation of dopamine and other catecholamines (e.g., norepinephrine). It is involved in the regulation of dopamine levels in the prefrontal cortex and genetic variations in the *COMT* gene have been associated with cognitive and emotional processing differences. Thus, altered *COMT* activity could potentially influence dopamine levels in brain regions relevant to cognitive functions and HANDs [37].

In turn, it is worth noting that *BDNF* supports the survival, growth, and maintenance of neurons; it is critical for promoting neuroplasticity, learning, and memory processes. *BDNF* is involved in various aspects of neuronal development and function and its dysregulation has been associated with several neurological disorders associated with HANDs. Indeed, the role of *BDNF* in promoting neuron survival and function makes it an essential factor to consider in the context of the pathogenesis of HANDs [37]. Furthermore, some studies have found an association between *ApoEε4* and HAD. This means that *ApoEε4* is of particular interest because it may be the key to understanding if HAD shares pathological mechanisms with AD and if HIV-infection can facilitate the development of AD [39].

Furthermore, genetic variants of several dopaminergic genes involved in dopamine pathways also appear to be implicated in different ways in the neurocognitive function of people living with HIV-1. Interestingly, an improvement in executive functioning in HIV-1-infected individuals is predicted by methamphetamine (Met/Met) genotype at the catechol-O-methyltransferase locus (rs4680), which is attenuated by methamphetamine abuse [40]. Moreover, *DRD3* (rs6280) was associated with neurocognitive illness only among people who abuse methamphetamine [40]. However, significant main and interaction effects for *DRD1* and *DRD2* have also been seen in cases of substance abuse for the ‘motor domain’ of neuropsychological performance in Caucasian populations and in nearly every neuropsychological domain in Afro-American patients [40]. Nonetheless, the GWAS conducted in the longitudinal Multicentre AIDS Cohort Study found no associations with *DRD2* or *DRD3,* although this work did not address interactions with substance or stimulant abuse.

A very elegant study [41] confirmed that downregulation of type-2 dopamine receptor mRNA in the prefrontal cortex was associated with more favourable neuropsychological and neuropathological outcomes. In turn, failure to downregulate type-2 dopamine receptor mRNA was significantly less favourable [41]. Methamphetamine affects the CNS by increasing the release and blocking the reuptake of dopamine, leading to elevated dopamine levels in the brain. Thus, the use of this drug by individuals living with HIV-1 exacerbates neurocognitive disorders because both the virus and the drug can impact dopamine-related pathways [42]. Indeed, the combination of HIV-1 infection and methamphetamine use may lead to further dysregulation of dopamine signalling, contributing to neurocognitive impairments in HANDs. Moreover, methamphetamine metabolism can generate oxidative stress and neurotoxicity byproducts that may cause additional damage to neurons in patients with HIV-1 [42]. Thus, the interplay between the effects of methamphetamine on dopamine and the neuroinflammatory response induced by HIV-1 infection can lead to more severe neurocognitive deficits in affected individuals [42]. Understanding these interactions is crucial for developing targeted therapies to address the neurological complications associated with both HIV-1 infection and methamphetamine use.

### 2.4. Drug Metabolism and Transport Genes

It is thought that many genetic differences in methamphetamine metabolism may be a factor in individual variation in the risk of neurocognitive impairment among HIV-infected methamphetamine abusers. Oxidative metabolism of methamphetamine requires the hepatic enzyme cytochrome P450, family 2, sub-family D, polypeptide 6 (*CYP2D6*). Therefore, the International Society of Psychiatric Genetics advises HLA-A and HLA-B testing before carbamazepine or oxcarbazepine are used. They also suggested that metabolisers based on polymorphisms in *CYP2C19* and *CYP2D6* are significantly more vulnerable to methamphetamine-associated neurocognitive disorders. These data are also supported by a recent study that included no HIV+ participants, thereby suggesting that genetic variation in *CYP2D6* and *CYP2C19* might be a risk factor for HANDs among HIV+ methamphetamine users [14].

### 2.5. Nuclear and Mitochondrial DNA Damage Genes

Nuclear and mitochondrial DNA (mtDNA) is thought to play a role in oxidative impairment. This damage can lead to neuronal apoptosis as part of the pathogenesis of HANDs first noted when cARTs were first being used, although involvement of this mechanism still remains unclear. During cART treatments, the production of neurotoxic HIV proteins such as Tat, gp120, and viral protein R (Vpr) is driven by the release of excitotoxic neurotoxins by macrophages and infected microglia, the increased presence of reactive oxygen species (ROS), and activated intracellular and mitochondrial calcium signalling pathways [43,44]. 

In this line, Zhang et al. compared the levels of nuclear and mtDNA oxidative damage (8-oxoG modification) in samples of postmortem frontal neocortex tissue from individuals with AIDS with or without HAND to seronegative controls [45]. This study revealed that the level of nuclear DNA 8-oxoG impairment was substantially higher in patients with HIV and HAND, with noncoding D-loop mutations in mtDNA being detected much higher frequency in HAND cases compared to both the other groups. This revealed the existence of a virus reservoir in the brain that produces oxidative damage and may contribute to the development of HANDs [45]. 

Another recent study in South African HIV+ women analysed the association of leukocyte telomere length and the presentation of HANDs in the context of chronic psychological stress caused by prior abuse and/or trauma [46]. This work reported that the activity of telomerase, the enzyme that synthesises telomeric DNA to prevent excessive telomeric DNA shortening, is reduced by oxidative stress, thereby producing short telomeres under chronic psychological stress conditions. The telomere lengths in these patients were significantly shorter compared to the controls, and this was strongly correlated to the severity of neurocognitive impairment. This study was very important because it highlighted, for the first time, the correlation between telomere length and neurocognitive function in patients with HIV-1 [46].

It is also important to highlight that the results of many ongoing studies analysing the role of mtDNA damage in promoting the onset and progression of neurodegenerative diseases remain controversial. For example, there is an urgent need for more research on Parkinson disease and amyotrophic lateral sclerosis diseases (ALS), which both involve mutated genes linked to mitochondrial function [46]. Many people with AD, Parkinson disease, or ALS, among others, have a ‘sporadic’ form of neurodegenerative disease, which is not directly associated with altered genes. These disease manifestations are characterised by alterations in energy metabolism which may suggest that they are mediated by alterations in mitochondrial function [46]. Importantly, many environmental, metabolic, and genomic risk factors act through mtDNA mutations [46], meaning that further investigation into the role of mtDNA damage and the molecular mechanisms associated with these diseases will be vital to eradicating them.

### 2.6. Matrix Metalloproteinase Genes

Matrix metalloproteinases (MMPs) are enzymes that play an important role in the remodelling and degradation of almost every component of the extracellular matrix. This degradation is of great importance because it is related to embryonic development and angiogenesis as well as cell repair and tissue remodelling. Altered MMP expression can generate abnormal extracellular matrix degradation which is the primary cause of the development of chronic degenerative diseases and other vascular complications.

Inside cells, MMPs are present in subcellular compartments in the endoplasmic reticulum, cytosol, nucleus, and mitochondria. The release of these enzymes increases cardiovascular problems, inflammation, kidney diseases, and malignant neoplasms, also increasing regulation of the innate immune response by exerting antiviral effects. In addition, they can also affect gene expression and transcription and, together with natural tissue inhibitors, they cause chronic inflammation, which is the basis of the pathogenesis of HAD [26].

Increased MMP expression and activity, together with imbalances in tissue inhibitors of metalloproteinases (TIMPs) in peripheral blood and CSF cells in patients with HIV-1 can cause damage and lead to increased permeability of the BBB by disrupting the junctions between endothelial cells and the extracellular matrix [47,48]. Together, these mechanisms increase the migration of infected and uninfected immune cells into the CNS and can produce HANDs. Of note, different studies have studied the implication of these genes in the development of HANDs [48,49]. For example, an *MMP-2* gene polymorphism has been implicated in the development of HAND diseases, both alone and synergistically with an *MMP-9* variation, in the peripheral blood leukocytes of both HIV-infected individuals and controls [50]. 

However, these data could not be replicated and so further research will be required to validate possible connections between MMP polymorphisms and HANDs. Additionally, the expression of *MMP* should also be studied given that its regulation seems to be involved in HANDs. Furthermore, new therapies and promising therapeutic options focussing on selective inhibition of intracellular MMPs could be useful for managing different pathological conditions.

**Table 1 ijms-24-14364-t001:** Summary of relevant genetic factors involved in HIV-associated neurocognitive disorders.

Function	Genes Involved	References
Neuronal injury, neuroinflammation, and neurocognitive degeneration	soluble CD14 (sCD14) [13,17,18] soluble CD163 (sCD163) [13] macrophage chemo-attractant protein-1 (MCP-1) [16] neopterin [13,16,21] neurofilament light chain protein (NFL) [17] tumor necrosis factor-α (TNF-α) [16] macrophage inflammatory protein 1 (MCP-1/CCL2) [16] macrophage inflammatory protein 1-α (MIP-1α/CCL3) [16] stromal cell-derived factor-1 (SDF1) [16] mannose-binding lectin 2 (MBL2) [6,29,30,31,32,33] inducible nitric oxide synthase (iNOS) [6] surface CD16/32 [6] interleukin-6 (IL-6) [6] nitric oxide NO [16] interleukin-15 (IL-15) [19] C-Reactive Protein (CRP) [20] C-X-C Motif Chemokine Ligand 10 (CXCL10) [21] T lymphocytes (CTLs) [25]. HLA class I and HLA class II genes [25,26,27,28]	[6] Bhatia et al. (2014) https://doi.org/10.7860/JCDR/2014/7725.4927 [13] Jumare et al. (2020) https://doi.org/10.1097/QAI.0000000000002320 [16] Silveira et al. (2022) https://doi.org/10.1002/cbf.3685 [17] Anderson et al. (2020) https://doi.org/10.1097/QAI.0000000000002484 [19] Yan, J et al. (2021) https://doi.org/10.1002/cti2.1318 [20] Abu-Rumeileh et al. (2019) https://doi.org/10.1186/s13195-019-0562-4 [21] Khalil et al. (2018) https://doi.org/10.1038/s41582-018-0058-z [25] Olivier et al. (2018) https://doi.org/10.3390/ijms19113594 [26] Sanna et al. (2017) http://doi.org/10.1371/journal.pone.0175316 [27] Siangphoe et al. (2015) http://doi.org/10.1097/QAI.0000000000000800 [28] Sagar et al. (2017) http://doi.org/10.1371/journal.pone.0181642 [29] Singh et al. (2008 http://doi.org/10.1016/j.jaci.2008.05.025 [30] van Rij et al. (1999) http://doi.org/10.1086/314940 [31] Levine et al. (2016) http://doi.org/10.1007/s13365-015-0410-7 [32] Singh et al. (2011) http://doi.org/10.2147/NBHIV.S19969 [33] Jia et al. (2017) http://doi.org/10.1002/ajmg.b.32530
Slower executive activity and memory and local release of serotonin in HIV+ patients	SNPs of serotonin-related genes: SNP rs4570625 [12,35] galactose mutarotase rs6741892 [35]	[12] Borrajo et al. (2021) https://doi.org/10.1080/07853890.2020.1814962 [35] Naranbhai et al. (2017) http://doi.org/10.1007/s00251-017-1000-z
Neurocognitive illness	Lipid metabolism: apolipoprotein Eε4 (ApoEε4) [40] Dopaminergic system: dopamine active transporter (DAT), Catechol-O-methyltransferase (COMT), Brain-derived neurotrophic factor (BDNF) [39,40] dopamine receptor D1, D2 and D3 (DRD1, DRD2, DRD3) [39,40,41] Immune system genes: C-C chemokine receptor type 5 and 2 (CCR5, CCR2) [44] C-C chemokine ligand type 2, 3 and 5 (CCL2, CCL3, CCL5) [44]	[39] Penedo et al. (2021). https://doi.org/10.1016/j.bbih.2021.100199 [40] Ojeda-Juárez et al. (2021 http://doi.org/10.3389/fmolb.2021.721954 [41] Gelman et al. (2012) https://doi.org/10.1007/s11481-012-9345-4 [35] [44] Borrajo et al. (2021). https://doi.org/10.3390/biomedicines9080925
Neurocognitive impairment by drug metabolism and transport genes	Metabolizers based on polymorphisms: enzyme cytochrome P450, family 2, sub-family D, polypeptide 6 (CYP2D6) [14] enzyme cytochrome P450, family 2, sub-family C, polypeptide 19 (CYP2C19) [14] Lipid metabolism: apolipoprotein E3 and E4 (ApoE3 ApoE4) [14]	[14] Nordström et al. (2013) http://doi.org/10.1001/jamainternmed.2013.9079
Oxidative impairment	Nuclear and mitochondrial DNA damage genes: nuclear DNA 8-oxoG [46]	[46] Malan-Müller et al. (2013) https://doi.org/10.1371/journal.pone.0058351
Extracellular Matrix remodelling leading to blood-brainbarrier permeability damage	Matrix metalloproteinases (MMPs): matrix metalloproteinases 2 and 9 (MMP2, MMP9) [47,48,49,50]	[47] Bazzani et al. (2022) http://doi.org/10.3390/ijms231911391 [48] Xing et al. (2017) http://doi.org/10.1016/j.bbi.2017.04.024 [49] Singh et al. (2018) http://doi.org/10.1111/apm.12817 [50] Singh H et al. (2016) http://doi.org/10.1002/jgm.2897

## 3. Transcriptomic and Epigenetic Studies

Astrocytes, neurons, and glial cells are all closely associated with neuropathogenesis in HANDs, especially HAD. Thus, numerous genes and biological pathways have been examined in genome-wide transcriptome studies to identify up- or downregulated activity in these cell types. Many of these studies have been replicated in simian and murine animal models but the use of homogenised brain tissue samples makes it difficult to study the molecular processes involved in the disease because they contain mRNA from numerous cell types [51].

Many studies have avoided this problem by performing gene expression analyses on specific brain regions such as the frontal lobe. Nonetheless, these regions are often not representative of the disease under study because they do not reflect the transcriptional changes that occur in other brain regions potentially affected in HAND. Another problem is that most studies use brain tissue samples from animals or humans that died at a very advanced disease stage (i.e., HIVE or HAD). This makes it difficult to know if these data can be extrapolated to other stages of HAND disease. 

Therefore, monocyte transcriptome analysis and comparative studies between HIV-infected monocytes and seronegative individuals have recently been carried out. Because of the key role of these cells in BBB damage, this work may be especially useful for in vitro microarray-based studies to identify upstream biological mechanisms relevant to early-stage HAND or HAD [52]. In this sense, one study using microarrays on brain tissues from HIV-infected and seronegative patients enrolled in the National NeuroAIDS Tissue Consortium Brain Bank showed two different transcriptome patterns in cases of HANDs versus HIVE and HAD [53].

It is also important to study genes and pathways up or downregulated in astrocytes, neurons, and glial cells (which are intimately involved in the pathogenesis of HAD); chemotaxis and inflammation-related genes; genes involved in the interferon (IFN) response; and genes that promote antioxidant and anti-inflammatory responses [53]. Dysregulation of genes involved in mitochondrial function, cancer, immune response, synaptic transmission, and cell–cell signalling have also been suggested as targets for further research in other studies on HAND [53].

Divergence between individuals with HIV-1 and cognitively normal patients with MND have also been found in around 1000 genes using RNA-seq to analyse micro RNA (miRNA), long non-coding RNA (lncRNA), and other factors [52]. However, there are some difficulties with this technique such as hybridisation background noise and saturation, incorrect probe annotation, poor isoform coverage, etc. The most upregulated genes between the groups were associated with viral replication and immune response, while the most downregulated genes were related to synaptic maintenance. In addition, the c/EBPβ transcription factor and most of its targets were altered in cases of MND, with most of these changes being astrocyte-specific [49]. Furthermore, transcriptome data interpretation based on biological systems methods can also help to uncover new therapeutic strategies and disease targets [52].

Current epigenetic research on HAND neuropathogenesis is exploring the involvement of miRNA pathways in infected tissues or cells. These pathways interact with mRNA to alter transcription and translation, providing further evidence for known neuropathogenic mechanisms such as elevated caspase-6 levels and mitochondrial dysfunction [54,55,56]. Previous works that used miRNA and gene expression assays in the context of HANDs have studied the impact of viral protein R (Vpr) in a human neuronal cell line, as well as the mechanisms underlying the altered expression of cytokines and inflammatory proteins in the CNS. The results showed that, via Vpr, a miRNA-dependent pathway produced neuronal dysfunction and dysregulation of the *CREB* gene, which itself is involved in miRNA and histone regulation [54].

Xu et al. studied 17 important altered miRNAs in neocortical brain tissues from patients with HANDs and compared them to samples from patients with HIV-1 without HANDs. Target genes related to peroxisome biogenesis and increased expression of immune genes such as *IFIT2*, *IFI16*, viperin, *IRF1*, and *OAS1* were found among the altered miRNAs [57]. In addition, changes in brain miRNA during HIV infection were also identified [57]. Another study examined the influence of Tat upon the expression of histone deacetylases (HDACs), principally HDAC2, which regulates genes involved in synaptic plasticity and neuronal function [58]. These researchers found that expression of the *CREB* and *CaMKIIa* genes (which are also involved, in neuronal regulation) was negatively correlated with *HDAC2* expression [58].

Nonetheless, few studies studying DNA methylation in the context of HANDs have been performed to date. However, a very noteworthy study in 17 HIV+ patients examined at two-time points suggested that DNA methylation levels could predict neurocognitive damage. Methylation profile correlation data at time point 1 suggested 26 strongly positively correlated autosomal sites and 18 negatively correlated sites. Correlation analysis between changes in the neuropsychological scalar score and methylation profile at time point 2 revealed 48 positively correlated and 26 negatively correlated autosomal sites, with the former indicating improved neuropsychological functioning and the latter, improvement in neurocognition in neurodegenerative disease [59].

## 4. Current Evidence for Different Therapeutic Approaches

The need to address the neurological complications caused by HAND requires the development of therapeutic therapies such as adjunctive neuroprotective approaches or FDA-approved drugs, among others. These pharmacological strategies include early cART, treatment simplification, treatment intensification, and reversal of HIV latency to preserve and improve sustained viral suppression in the brain. Of these, adjunctive neuroprotective therapies, FDA-approved drugs, cognitive therapy, and aerobic exercise combined with improved cART offer plausible strategies for optimising the prevention and treatment of HANDs.

### 4.1. Antioxidative Drugs

Dimethyl fumarate (DMF) is an antioxidative drug that modifies the kelch-like ECH-associated protein 1 pathway, leading to upregulation of antioxidant genes [60]. DMF also inhibits the nuclear factor B (NF-kB) pathway, with NF-kB activation acting as a start signal for HIV-1 transcription. Thus, blocking the ability of HIV-1 to sense immune cell activation could help prevent neurodegeneration. DMF induces antioxidant responses, suppresses HIV replication, and reduces the release of neurotoxins from infected cells in an HIV+ monocyte-derived macrophage model [61] (Figure 3). However, there is an important limitation to this drug because it produces apoptosis in distinct T-cell subsets. Therefore, because of the risk of progressive multifocal lymphocytopaenia, patients should be closely monitored if they receive this type of treatment in a clinical trial.

### 4.2. Psychiatric Medications

Paroxetine is a selective serotonin reuptake inhibitor. In 2014, Meulendyke et al. found that both paroxetine alone and in combination with fluconazole (an oral anti-fungal medication) provided neuroprotection against gp120 and Tat neurotoxicity [62] (Figure 3). This study showed that daily oral treatment with combined fluconazole and paroxetine in simian immunodeficiency virus-infected rhesus macaques had favourable neuroprotective effects but did not reduce cellular stress, levels of inflammatory markers such as CCL2 and IL-6 in the CNS, or inhibit viral replication in the brain, CSF, or peripheral nervous system [62]. 

Both paroxetine and fluconazole are already approved for human use and their combination is used as placebo to attenuate HAD. Patients receiving paroxetine showed improved cognition that helped with daily functions such as verbal fluency, visual attention, and task switching [62]. In turn, fluconazole treatment did not improve any cellular stress markers or cognition [63]. In summary, studies on the role of paroxetine in HAD therapies must be deepened because this drug is well tolerated in humans and can improve some cognitive functions in patients with HIV-1 [63].

### 4.3. Anti-Inflammatory Drugs

Among the different anti-inflammatory drugs available on the market, it is worth highlighting minocycline and meloxicam. Minocycline is an alternative treatment to inflammation in cases of HAD. Based on in vitro models of brain inflammation, this drug is considered a potent microglia activation inhibitor with neuroprotective activity by reducing apoptosis, the release of pro-inflammatory cytokines, and nitric oxide production [64]. Coupled with its highly lipophilic profile, minocycline can easily cross the BBB [64], making it a promising candidate for HAND therapy. There is evidence to support the anti-inflammatory effects of this drug in a previous work in which intracerebroventricularly administering minocycline to gp120 rats reduced the levels of protein and mRNA inflammatory markers [65].

In turn, meloxicam is a nonsteroidal anti-inflammatory drug that reduces prostaglandin production, and thus inflammation, by selectively inhibiting COX-2 enzymes. This drug is employed both in veterinary and human medicine and has been approved by the FDA for the treatment of osteoarthritis. Interestingly, treatment with meloxicam can reduce CCL2 expression, which is increased by HIV-1 proteins. This may be important because increased CCL2 was associated with depression-like behaviours in rats caused by alterations in neuroinflammation and cell proliferation [65]. However, this work showed that meloxicam was unable to attenuate depressive behaviours, and so the potential role of meloxicam in HAD therapy remains unclear (Figure 3).

### 4.4. Antiretroviral Drugs

Maraviroc is an antiretroviral that acts as a CCR5 antagonist by blocking this receptor, thereby preventing HIV-1 virus entry into target cells [66,67] (Figure 3). This drug shows poor CNS penetration and intra-brain distribution but low neurotoxicity [66] and some enrichment in gut-associated lymphocytic tissue (GALT). Given that endotoxins can accumulate in the intestine in cases of HAD, the characteristics of maraviroc make it a good alternative treatment or preventive measure for the disease by protecting GALT [66]. Previous in vitro studies have shown that maraviroc can inhibit the migration of macrophages in response to CCL2 [66,67] to produce anti-inflammatory response. However, new clinical studies with more sensitive assessments and larger sample sizes must be designed to detect changes in inflammatory markers and to evaluate the therapeutic potential of maraviroc.

### 4.5. Interferons

Interferon alpha (IFNα) and interferon beta IFNβ are type-I IFN protein cytokines secreted by host immune cells to inhibit viral replication, although these remain unregulated during early infection stages. Different studies have demonstrated that type-I IFNs are involved in both attenuating and exacerbating neuroinflammation caused by HIV-1 infection [68] (Figure 3). In recent work, in vitro experiments on rat cerebrocortical cells showed increased expression of several chemokines (CCL5, CCL4, CCL3, and CXCL10) after IFNβ treatment, revealing that CCL4 is responsible for mediating the neuroprotective effects of IFNβ [68]. 

In turn, in vitro experiments in gp120 transgenic mice treated with intranasal mIFNβ showed that by restricting HIV-1 infection and brain pathogenesis, IFNβ increased CCL4 mRNA expression, MAP-2 and synaptophysin levels, and decreased Iba1 levels [68]. This means that IFNβ is now a promising therapeutical option in the context of HAD, but behavioural studies in this respect are still lacking. Koneru et al. also showed that IFNα had neuroprotective effects in a mouse model treated subcutaneously with B18R (an IFN receptor that can inhibit type-I IFNs in many species), which, in combination with a common cART regimen (atazanavir, tenofovir, and emtricitabine), prevented astrogliosis. Thus, despite absence of behavioural studies, this therapy may still be promising for the treatment of HAD [69].

Nevertheless, another study has shown that chronic activation can drive immune exhaustion, with type-I interferons (IFN-I) emerging as critical components underlying ongoing activation in HIV infection. These authors tested the effect of blocking IFN-I signalling on T cell responses and virus replication in a murine model of chronic HIV infection. They showed that the in vivo blockade of IFN-I signalling during chronic HIV infection diminished HIV-driven immune activation, decreased T cell exhaustion marker expression, restored HIV-specific CD8 T cell function, and led to decreased viral replication. Furthermore, in combination with IFN-I blockade, cART accelerated viral suppression, further decreased viral loads, and reduced the persistently infected HIV reservoir compared with cART treatment alone. Together, these data suggest that blocking IFN-I signalling in conjunction with cART treatment can restore immune function and may reduce viral reservoirs during chronic HIV infection, thereby validating IFN-I blockade as a potential therapeutic route for treating HANDs [70].

## 5. Conclusions

cART regimens have helped millions of people with HIV-1, facilitating the management of the infection. However, despite the efficiency of cART, neurological disorders such as HAD (the most severe form of HAND) still persist in many patients living with HIV-1. The diagnosis of HAD requires exhaustive testing, and several genetic and environmental factors can be associated with its persistence. Some of these factors are related to continuous brain inflammation caused by low-level HIV-1 replication in cellular reservoirs such as microglia [71], the release of viral proteins, and poor antiretroviral drug penetration across the BBB. Thus, there is still a pressing need for an efficient therapeutic strategy to predict HAD pharmacokinetics in silico, followed by comprehensive in vivo studies of novel cART and anti-reservoir and other anti-host factor drugs.

Current therapies and potential future treatment approaches are dependant on clinical trials to assess new therapies and novel therapeutic approaches to existing drugs. The presence of cellular RNA and DNA in CSF has also been explored as potential surrogate markers for neurocognitive impairment. While these markers are detectable in people living with HIV-1 on cART, it is uncertain if changes in these markers correlate with clinical improvement. In addition, certain HIV proteins such as Tat, gp120, Vpr, and Nef have been identified as neurotoxic. Thus, novel antiretroviral strategies could potentially target these proteins, although practical implementation will be challenging.

Ongoing research has resulted in promising treatment strategies which include demonstration of the effectiveness of IFNβ and maraviroc. However, further research will still be necessary to examine and analyse which pharmacological therapeutic strategies that can be used in clinical settings. Furthermore, improved CNS delivery systems, such as nanodiamond or nanogel formulations, could perhaps enhance the effectiveness of cART in reducing HIV brain loads. Other techniques based on preventing HIV-infected cells from entering the brain could reduce the CNS HIV load, but there are still concerns regarding immune surveillance and potential adverse effects.

Various novel treatments addressing the pathogenic mechanisms of HIV-induced brain disease are currently under investigation, including PPAR-gamma agonists, PD1 inhibition, curcumin, intranasal insulin, and Jak inhibitors. Ongoing research includes a phase II clinical trial investigating the effects of baricitinib on the persistence of HIV in the CNS and exploring potential treatment options that address neuroinflammation and cognitive impairment. Finally, it is important to highlight the persistence of the neurocognitive impairment in people with HIV-1, the challenges in clinical trials, and the potential for new treatments to address the complex interplay between HIV infection, neurotoxicity, and cognitive decline.

## Figures and Tables

**Figure 1 ijms-24-14364-f001:**
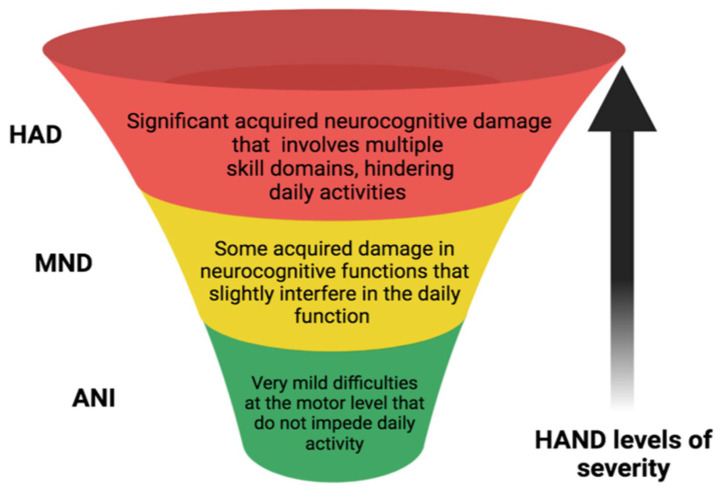
Schematic diagram which presents comparisons of the Frascati Criteria for diagnosing HAND.

**Figure 2 ijms-24-14364-f002:**
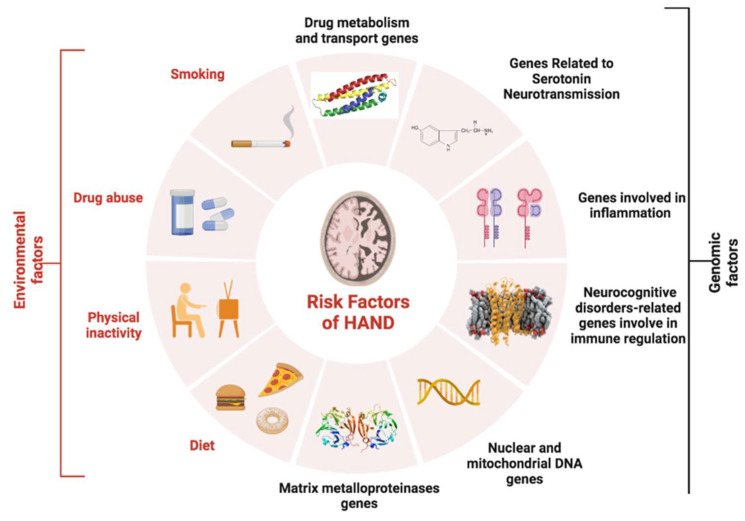
Illustration showing the different environmental and genomic risk factors involved in the appearance of HIV-associated neurocognitive disorders (HANDs).

**Figure 3 ijms-24-14364-f003:**
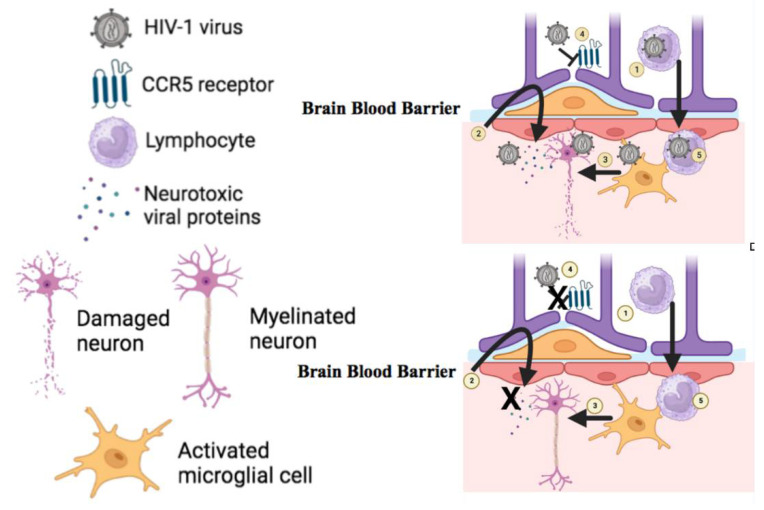
Representation of the current diverse therapeutic approaches for HAND neuropathogenesis. (1) Dimethyl fumarate (DMF) inhibits the nuclear factor β (NF-kβ) pathway, inhibiting HIV replication. (2) Psychiatric medications provide neuroprotection to neurotoxic HIV-1 particles (3) Anti-inflammatory drugs downregulate pro-inflammatory cytokines, preventing neuronal damage. (4) Maraviroc acts as a CCR5 antagonist and blocks the receptor to impede HIV-1 entry into target cells. (5) Interferons inhibit viral replication of HIV-1 viruses.

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
