# Peer review of "Genomic Factors and Therapeutic Approaches in HIV-Associated Neurocognitive Disorders: A Comprehensive Review"

_ijms, 2023, doi:10.3390/ijms241814364_

Round 1

Reviewer 1 Report

In this review article, Ana Borrajo et al. summarized the current data on the epidemiologic, genetic, transcriptomic, and epigenetic studies of HAND. They also discussed the pharmacological/therapeutic strategies for HAND. The description of HAND epidemiology is quite good while some of the other section needs to be improved, starting from 2.2.

Major points:

1.  In 2.2, please have a brief discussion of serotonin signaling and its role in normal neurobiology and HAND. Many HIV researchers may not know too much about it.

2. Similarly, in 2.3-2.4, please briefly discuss the functions of DAT, DRD2/3, COMT, and BDNF and the metabolism of methamphetamine.

3. In 2.5, it was not clear which cell types were studied for DNA damage/8-oxoG. If it was not in neuronal cells, it may be not relevant to HAND.

4. Similarly, in 2.6, which cell types were studied regarding MMP2?

5. A significant part of this review is about therapeutic approaches to HAND, this needs to be reflected in the title. Please revise the title.

6. Figure 3, needs to be revised. It should have two parts: one for the condition during HIV infection and the other for therapeutic condition/ART. Otherwise, it is confusing to read.

7. In 4.1, NF-kβ probably is wrong, it should be NF-κB. Please make sure this is correct.

8. From 4.3, there are many typos, please proofread the sections from there.

9. In line 558, “FDA approved for treat” should read “approved by FDA for the treatment of”.

10. In 4.4, “poor distribution to the CNS penetration” is confusing, are you talking about “poor distribution” or “poor penetration”?

11. In 4.5, please note that IFN-I activation is highly related to stable HIV reservoir and impaired immune responses during chronic HIV infection receiving effective ART. Two JCI papers reported this in the Hu-mice model. So, I do not believe IFN therapy will work. Instead, anti-IFN-I may be beneficial for reducing viral reservoirs, CD4 recovery, and possibly HAND. This section needs to be discussed in detail.

12. In line 598, CART is weird. You may just use ART or cART.

13. Please cite a recent JCI paper regarding HIV reservoir in microglia: doi: 10.1172/JCI167417, which comprehensively characterize microglia as true HIV reservoirs in the brain of PWH on ART.

14. In line 606, antiretroviral drugs are not accurate here. It probably includes ART plus anti-reservoir or anti-latency, and other anti-host factors drugs.

The quality of English language can be further improved.

Reviewer 2 Report

Genomic Factors and Their Impact on HIV-Associated Neurocognitive Disorders: A Comprehensive Review 

Line 23: “life expectancy and reduction..” - can use Oxford comma instead of and

Line 24: “The pathological mechanisms contributing to HAND..” - Recommend splitting the sentence into two.

Line 28: “Throughout the years, multiple pharmacological approaches..”  - Recommend splitting the sentence into two.

Abstract: Needs to be include a sentence or two about why genetic studies in HAND are important and why this review aims to focus on them. 

Line 40: Expand HIV.

Line 42: “The incidence of these…” - sentence unclear.  Maybe the author meant: The incidence peaks at young adulthood with increase in late adolescence. ?

Line 44: “Mental health disorders..” - sentence restructuring required. Replace duty with role. 

Line 45: “People with serious mental illness..” -sentence restructuring required.

Line 47: “multiple sexual partner..” - Oxford comma after the word partner.

Line 50: What is HIV-1 ? Define here if different than the normal definition of HIV.

Line 49: “The risk of HIV infection may also increase with the severity of psychiatric disease.” - unclear how the following examples explain this observation. The examples listed showcase a higher incidence of anxiety and depression in HIV positive population but do not imply reverse correlation. The presence of anxiety and depression does not make the population more susceptible to getting HIV. 

Line 54: “In 2018..” - Oxford comma after 2018.

Line 58: “What is more,..” - replace with Furthermore.

Line 60: “a study made in Spain..” - remove made.

Line 63: “adherence to pre-expose prophylaxis (PrEP)” - Oxford comma after bracket

Line 67: “which also impact..” - should be “impacts”.

Line 73: “as well as other” - replace with “ as one of the”

Line 75: “host immune responses, inflammation, and nervous system cells in HAND.” - replace with “host immune response, inflammation, and cells of the nervous system in HAND”

Line 78: Did the authors mean to indicate the identification or standardization of the HAND phenotypes is challenging because ….. ? The complexity of research etc.. has no bearing on the actual manifestation of the phenotypes observed in HAND. Sentence needs restructuring. 

Line 82: Better to move this to the top of the section 1.1 

Line 88: “cognitive-functional” - can remove functional and just say “cognitive dysfunction”.

Line 89: “The Frascati criteria require an affection of at least two cognitive areas to 89 diagnose a HAND.” - sentence restructuring required.

Line 92 - 100: “ In the Frascati criteria, … (Figure 1)” - check grammar and sentence structuring. 

Figure 1 : suggest to distill the words into bullet points or short sentences in the figure. The text descriptions are too lengthy. Can this be represented as a table ? 

Line 105: “Many of the affected patients do 105 not receive the necessary pharmacological therapy..” - contradicts with Line 108 which indicates the necessity for genetic studies. Is the higher failure rate of HIV therapy on the low- and middle-income counties stemming from incomplete treatment plans or lack of treatment efficacy ? This section needs to clarify that distinction clearly. 

Line 120: “Moreover, it has been found.. “ - can remove “it has been found” and just state that a significant association between HAD and mental illness such as… is present or exists. 

Line 125: replace affect with affects

Line 128: “Generally, there is an..” - is lack of self confidence the only mental indicator in HAND ? Sentence is not clear and needs to be reworded. 

Line 130: “As a result, HAND (predominantly HAD) patients..” - is the conclusion that the patients with severe mental illness and HAD are not consistent with their medication or that they overuse them ? It is not clear from the sentence and revision is requested. 

Line 137: “Psychiatric disorders such as schizophrenia or bipolar disorder are frequently found in individuals living with HIV who abuse substances and come from disadvantaged social and economic backgrounds.” -  what is the rate of incidence of co-occurrence of the psychiatric disorders and HAND in patients from disadvantaged social and economic background compared to control population ? 

Line 157: “This review describes findings from studies investigating the role of human host genetic factors in the pathogenesis of HAD..” - the introduction so far has focussed on the co-occurrence of other mental illness as well as current therapeutics without any mention of genetic factors at play or the need to  study them. It would be better to add additional context or revise existing content to connect the observed phenotype to variations in genetic factors etc. 

Line 172: Figure 2 description should say schematic or illustration instead of graph. 

Line 183: “(MCP-1/CCL2)” - this should not be in brackets and should be followed by an Oxford comma.

Line 201: “What is more..” - replace with Furthermore

Line 214: “diagnosticated” - this is not a valid word. Replace with “ used in diagnosis of …”

Line 223: “NFL,..” - comma not necessary

Line 235: “viral cells” - Oxford comma after cells

Line 246: “Even more, ..” - remove even more from sentence

Line 252: “Subsequent work postulated that the HLA-DR genotype..” - HLA-DR is a broader term for the HLA-DR locus. Does this imply none of the alleles within this locus were important ? This contradicts statements made in line 242 which implicate HLA-DR*04 with increased risk of HAD. Request authors to clarify.

Line 307: “Also, it has been observed an effect of the CCR2 genotype and the genotype of one of the natural ligands of 308 CCR5, CCL3, on HIV-associated neurocognitive disease” - request the effect of this genotype be explained. Is is neuroprotective ?

Line 333: “meth” - avoid short forms or alias if not specified earlier in the text. 

Line 347: “transcriptomic analyzes” - perhaps it should read “transcriptomic analytes” ?

Line 341- 363 - Unsure how the explanation of effect of Cadmium exposure and ApoE interaction is directly relevant to HAND. The explanation in these paragraphs is about AD specifically without explaining the rationale or association to HAND/HAD. Suggest revision of these paragraphs to bring the focus into HAND. 

Line 365: replace has with have 

Line 369:  replace with “activated intracellular…..pathways.”

Line 383: replace with “produces short telomeres under chronic psychological stress conditions.”

Line 400: “Because of this, it is of vital importance to investigate the role of mtDNA damage and the molecular mechanisms associated with these diseases to eradicate them.” - Can the authors provide some suggestions for how to accomplish this task ? Or provide some framework for how they would accomplish this task ?

Line 406: “These enzymes are responsible for increasing cardiovascular problems, inflammation, kidney diseases and malignant neoplasms, also increasing the regulation of the innate immune 408 response by exerting antiviral effects.”  - did the authors mean in case of a dysfunction ? The MMPs on their own do not cause the above problems but need to be triggered to do so. The above sentence needs to better clarify that information.

Line 412: “imbalances with TIMPs..” - expand TIMPs

Line 418: “there is a study that..” - remove wording

Line 426: “the focus” - replace with “focussing on”

Table 1 - please format the table to have equal spacing and same font size and style

Line 486: “It is found, changes in brain miRNA during HIV infection.” - sentence requires restructuring. 

Line 499: “improvement neurocognitive” - replace with “improvement in neurocognition” 

Line 580: “works” - replace with studies

Line 583: “investigated in vitro and..” - add with after investigated

Line 584: Missing In at the start of this sentence “In vitro experiments..”

Line 587: “Data from gp120 transgenic mice experiments treated with..” - replace with “Experimental data from gp120 transgenic mice treated with..”

In the conclusions section , the authors should tie in the original goals of the review which was about understanding genetic factors at play in HAND with the current therapeutic landscape and future directions. 

Significant editing required at places, along with grammar check and replacing repetitive use of certain phrases. Also suggest to split complex lengthy sentences into two or more shorter sentences. Ensure there is a correct flow of information between paragraphs. 
